# Effect of Relationship Quality in Collaboration and Innovation of Agricultural Service Supply Chain under Omni-Channel Model

**Baojun Yang [1], Bo Yuan [1], Ning Yang [1], Yan Liu [1], Ruiqi Jia [1], Yongyan Wang [1], Ting Miao [1,\*], Jianxu Liu [2,3] and Songsak Sriboonchitta [3]**

1. School of Business, North Minzu University, Yinchuan 750021, China
2. School of Economics, Shandong University of Finance and Economics, Jinan 250014, China
3. Faculty of Economics, Chiang Mai University, Chiang Mai 52000, Thailand
\* Correspondence: miaoting@nun.edu.cn

**Abstract:** The goal of this study is to investigate the regulation effect of relationship quality in the process of the omni-channel (OC) model on service supply chain (SSC) collaboration of agricultural products. Furthermore, it is also to explore the intermediary effect of SSC collaboration in the process of service innovation and OC. A questionnaire was developed, research data were gathered from businesses in the agricultural SSC in western China, and an empirical study was carried out by using the AMOS multivariate statistical analysis approach after a thorough review of the literature in recent years. The study demonstrates that the OC model has a considerable impact on service innovation, SSC collaboration has an intermediary effect, and the quality of supply chain (SC) relationships has a regulation effect in the model. The results inspire academics and industry professionals to focus on SSC collaboration, improve the OC model's administration, and promote service innovation in agricultural SC. Finally, the paper proposes suggestions to promote agricultural product development in western China in terms of enhancing SSC collaboration, OC model, and service innovation.

**Keywords:** omni-channel (OC) model; service supply chain (SSC); agricultural products; service innovation; China

## 1. Introduction

In recent years, two types of SC systems have emerged in the field of SSC management, namely service-only supply chains (SOSC) and product–service supply chains (PSSC) [1], with PSSC receiving increasing attention. The SC system serving product sales enhances the service level of products, and Stank et al., argues that the collaboration with external SC entities will improve the service performance [2]. SSC collaboration refers to the process of information sharing and mutual collaboration among SC entities to improve service efficiency. The agricultural SSC is one of the PSSC, which consists of producers, processors, wholesalers, service intermediaries, and retailers. Agricultural SSC collaboration connects all subjects through information sharing and mutual cooperation to achieve service delivery and SC operation efficiency improvement. Under the background of "Internet+", the operation model of agricultural SSC is facing transformation, and the way of product information transmission, online and offline operation channels has contributed to the collaboration of agricultural SSC. Especially in the OC model, with the assistance of modern information technology, SSC collaboration brings information sharing and confidence in order to improve service performance.

Since 2017, western China has implemented a rural revitalization policy, which has accelerated the development of e-commerce and logistics for agricultural products and improved operational efficiency by promoting the integration of online and offline channels. In the OC model, the agricultural PSSC is very important. In terms of research on the

relationship between agricultural products and SSC, Chamhuri and Batt studied the store-selection behavior of customers when purchasing agricultural products and found that customers who purchased in modern retail stores placed more importance on services such as convenience and entertainment [3]. Bandinelli et al. [4] examined the acceptance of Near Field Communication (NFC) technology by clients in the wine sector and concluded that Perceived security (PS) and Perceived compatibility (PC) were the keys to the choice. DAN (2018) analyzed the integration of agricultural products and life-type services in terms of response time, perceived quality, service process, and SC member collaboration from the perspective of Internet + fresh agricultural products and proposed a SC relationship between agricultural products and services [5]. Because of the perishability and the short transportation cycle of agricultural products, a number of researchers have conducted relevant researches to promote cold-chain transportation, logistical distribution and supply chain synergy.

In the survey of agricultural product service supply chain collaboration, service supply chain entities such as agricultural product manufacturers, suppliers, retailers and logistics providers need to trust and cooperate, and partnerships affect the performance of agricultural product service supply chain. Agribusinesses are gradually introducing the SSC management model [6] and are focusing on synergistic optimization. Unlike industrial products, the success of agricultural marketing requires not only the guarantee of production, but also the joint efforts of other parts, such as logistics and sales, to form a pattern of coexistence and co-prosperity, therefore, Matopoulos et al. [7] confirmed that SC collaboration is of great significance to the agri-food industry. The core issue of agricultural SSC collaboration is to integrate service resources and create customer value together, and to closely integrate customer value management, service process and capacity management through interaction and coordination among multiple participating subjects such as service integrators, service providers and customers. In the study of the impact of OC model on the agricultural SSC, Chiang et al. [8] proposed that the introduction of direct sales by manufacturers increases the overall profitability, thus analyzing the impact on the SC. Chen et al. [9] further analyzed the problem of manufacturers managing their direct online sales channels and independently owned physical retail channels when the channels compete on services problem. Dual-channel and OC become key influences in analyzing the SC structure. Wang and Tong take the agricultural PSSC to meet consumer demand through OC standardization and modernization of operations in the context of Internet development [10]. The OC model becomes an important direction to promote service innovation in the agricultural PSSC.

Many authors have conducted numerous empirical studies on SSC collaboration, but it is difficult to draw consistent conclusions because of differences among the studies and their collected sample data. It is generally believed that the purpose of SC collaboration is to reduce costs to enhance firm performance [11,12]. As a result, firms introduce the OC model [13–15]. In the era of digital economy transformation, factors affecting SC collaboration include information sharing [16,17], SC partnership and trust [11,18]. Thus, in-depth analysis of SC relationship quality is critical to improve performance. Meanwhile, the convergence of offline channels and new digital channels (e.g., mobile channels and social media) has facilitated the development of OC model, which places higher demands on SC collaboration. Niranjan et al. [19] analyzed the impact of the dynamics of customer choice on the green SC structure in the OC model through a study of the SC in the Indian market. Qi [20] analyzed the relationship between SC collaboration and performance from an OC perspective. OC retailers integrate digital and physical channels, providing offline and online information to deliver a superior shopping experience to customers. This model is fundamentally changing the way retailing is done and is triggering more and more manufacturers to reconsider their SC distribution strategies [13]. Therefore, it is important to explore the impact of the OC model on SSC collaboration.

At present, the overall slow development of agricultural products in western China is mainly limited by the SSC collaboration relationship and Internet technology. In the context

of Big Data and A.I. Technology, production and services are becoming highly digitalized, networked and computerized. It increases the trust and the perceived quality between producers and consumers.

Thus, the research objectives of this paper are to (1) study the influence of OC model on PSSC collaboration; (2) study the influence of OC model on service innovation; (3) study the problem of intermediary effect of SSC collaboration in the process of service innovation; and (4) study the influence of interrelationship between SSC subjects on service innovation.

To this end, this paper constructs a novel model to explore the impact of the OC model on service innovation and the intermediary effect of SSC collaboration. The model will explain the role of SSC collaboration in the process of adopting the OC model to achieve service innovation. Another contribution of this paper is the introduction of relationship quality. The moderating effect of partnership on SSC collaboration is analyzed using the relationship quality measures of trust, commitment, adaptation, common and collaboration. The model will provide important ideas for agricultural service innovation in western China.

The rest of the paper is organized as follows: Section 2 is the literature review. Section 3 presents the methodology, including sample selection and data sources, data reliability and validity analysis and correlation analysis of variables. Section 4 presents the results and discussions, including analysis of model fitting effect, analysis of hypothesis results and discussion of study results. Our last section summarizes the main conclusions and discusses the implications of our research.

## 2. Literature Review

### 2.1. Research Hypothesis

#### 2.1.1. OC Model and SSC Collaboration

In the context of the vigorous development of digital economy and big data technology, the construction of OC model is the key to promote the coordinated development of enterprise SCC. Liu et al. [21] explored the choice strategy of retail enterprises for channel structure and suggested that it is important to build an OC of SC collaboration strategy in the evolution from DC to OC. Pereira and Frazzon [22] studied the optimal combination between demand forecast and enterprise operation plan in retail SC from the perspective of methodology and proposed to realize the synchronous coordination of supply and demand in enterprise OC retail mode with data drive. Agatz et al. [23] compared the supply chain management (SCM) of DC and OC of the internet, and concluded that, compared with DC, OC model can better use economies of scale to create more efficient collaboration for SC management and serve different customer groups on the Internet.

Therefore, companies promote information sharing among cooperative enterprises by developing online and offline OC models. The synergistic operation efficiency of the SSC is improved through the joint decision making of each subject to achieve mutual benefits and win-win situation. This leads to hypothesis 1:

**H1:** *OC model has a positive impact on SSC collaboration.*

#### 2.1.2. OC Model and Service Innovation

OC model, as a new business model based on modern digital technology, can effectively improve the consumer experience, create value-added services, and promote the innovation of business model (Shi [24]). The survey of channel companies shows that promoting service innovation requires every member of the chain to play its own unique role. Capriello and Riboldazzi [25] proposed that tourism agency service innovation included retail channel integration, diversified retail marketing and customer relations, and analyzed that OC model has a significant impact on its network service innovation through case study. Cao and Li [26] suggested that, based on the OC model, the SC information technology of the U.S. retail industry promoted channel integration, and improved the ability of service innovation. Zhang et al. [27] explored the pre-sale strategy of fresh agricultural products SC from the perspective of the relationship between OC model and service innovation.

By establishing a perfect OC sales model, enterprises and SC partners can quickly automatically distribute orders according to consumer needs. This can further enhance supply chain management (SCM) quality, promote the development of core competitiveness, improve market share, and realize enterprise value innovation. Therefore:

**H2:** *OC model has a positive impact on service innovation.*

### 2.1.3. SSC Collaboration and Service Innovation

With the increasingly significant role of SSC in enterprise operation, enterprises began to pay attention to the integration of procurement and SC, and promote service innovation through SSC coordination, in order to improve competitiveness. Agarwal and Selen [28] pointed out that service innovation is multidimensional, including not only the formal innovation of products, but also the improvement of business performance and production efficiency. Stank et al. [2] conducted an empirical study on logistics services and proposed that, the coordination of external SC entities could improve the internal cooperation of enterprises, thereby improving the performance of logistics services. Soosay et al. [29] and Liao et al. [30] found that the synergistic relationship between enterprises and SC partners can integrate and connect their operation management, improve the efficiency of SC operations, and then directly or indirectly stimulate the innovation ability of enterprises. Li et al. [31] investigated the impact of organizational learning on service innovation performance, using SC collaboration as an intermediary variable, and the empirical study found that knowledge absorption and knowledge integration had a significantly higher impact on service innovation performance through the intermediary role of SC collaboration.

From the perspective of SSC collaboration and service innovation, the SC innovation and development of enterprises need not only the guarantee of production links, but also the joint efforts of SC links such as logistics, transportation, circulation and sales. It enhances the interaction and coordination of all participants in the SSC of enterprises, improves the efficiency of cooperation, and promotes innovative development through the integration and use of resources among SC enterprises. Overall, the hypothesis 3 is:

**H3:** *SSC collaboration has a positive impact on service innovation.*

### 2.1.4. The Role of Relationship Quality Regulation in SSC

The successful enterprises SC management stems from the good quality of SC relations. The good relationship and trust between enterprises and partners promote long-term cooperation between enterprises [32]. Fynes et al. [33] used the survey data of 200 electronic suppliers in Ireland to study the impact of SC relationship quality on the quality performance by dividing it into trust, commitment, adaptation, collaboration and other dimensions. Foo et al. [34] studies the quality of SC relationships and proposes that under the OC model, suppliers can better obtain the demand of consumers. Through a survey of 311 manufacturing enterprises in Western China, Su et al. concluded that the good cooperation between manufacturing enterprises and upstream and downstream enterprises in the SC promoted their collaborative development [35]. Kühne et al. [36] and Kim et al. [37] proposed that the relationship quality between suppliers and customers has a significant impact on the overall SC management capability, and harmonious relationship quality is an important prerequisite for improving the collaborative innovation in the supply.

Therefore, good relationships between companies and their partners facilitate them to form strategic alliances. This will enable them to obtain market information quickly and meet consumers' needs more accurately. Furthermore, they can strengthen synergistic relationships through cooperation to achieve OC model development. Thus, the hypothesis 4 is as follows:

**H4:** *SSC relationship quality positively regulates the effect of OC model on SSC collaboration.*

In the context of the digital economy, the OC model promotes service innovation, and enterprises need to strengthen SSC collaboration to promote service innovation. Thus, according to the research hypothesis, this paper constructs a model, as shown in Figure 1:

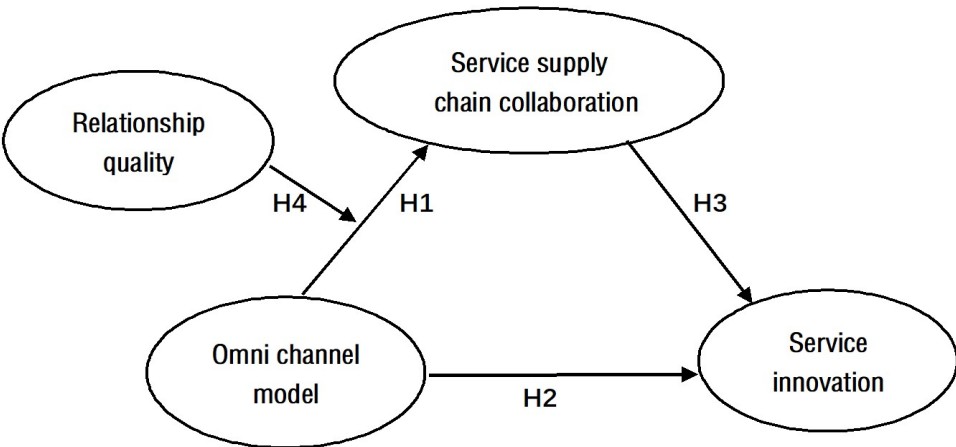

**Figure 1.** OC model, SSC collaboration and service innovation model.

### 2.2. Variable Definition and Measurement

This study was based on the maturity scale, combined with the reality of the survey and the respondents' understanding of the problem. The academic community has formed a relatively mature scale to measure the relationship between variables when studying the OC model, SC collaboration, service innovation and other issues. Among them, OC model variables refer to Hüseyinoğlu et al. [38]. The core of SSC collaboration is whether members can realize revenue sharing, information sharing and abide by the spirit of contract, the scale refers to the research results of Stank [2].

The scale of service innovation, including the innovation of service concept, service technology, service process and service content, refers to and has been modified according to Grawe et al. [39]. Research shows that trust, commitment and relationship atmosphere are important factors to measure the quality of SC relationships [40,41]. Trust is the basis for cooperation and facilitates transactions between parties; commitment is the key to stable relationship development and makes it easier for both parties to manage the work; and a good relationship atmosphere promotes closer cooperation and better SC operations. Based on the research of Fynes et al. [33] and He [42], the questionnaire was designed. Data were measured by a five-point Likert scale method, and the respondents choose to score between '1–5' according to their actual experience and the actual situation of the enterprise.

While conducting the study and considering that the respondents including enterprises and farmers, a total of 50 questionnaires were distributed. The questionnaires including 40 graduate and undergraduate students, especially 5 farmers and 5 enterprise representatives. The pre-survey involves two main issues: Firstly, it needs to examine the respondents' understanding of the questionnaire questions and verify that the questionnaire met the intention of the investigator; secondly, it is important to test whether there was any deviations in understanding between the individual farmers and the enterprise representatives. During the pre-survey process, the respondents proposed changes to the questions and improved the topics. After testing the individual farmers and enterprise representatives, it was found that their understanding of the questionnaire items was consistent and without understanding bias, which have solved the problem of homogeneity. Thus, finally the questionnaire of this topic has been formed.

## 3. Research Methodology

### 3.1. Sample Selection and Data Source

The group selected the Chinese wolfberry (agricultural product) production, distribution and logistics service providers in the western region of China as the research object. Stratified random sampling method was adopted to issue questionnaires, 350 questionnaires were issued, excluding some questionnaires with incomplete data, 275 questionnaires were recovered, and the effective recovery rate was 78.6%. with the detailed statistics of the characteristics listed in Table 1.

**Table 1.** Demographic characteristics of the sample.

| Statistical Variables | Measurement Items | Frequency | Percentage |
|---|---|---|---|
| Operating years | below 5 years | 118 | 42.9 |
| | 5–10 years | 89 | 32.4 |
| | above 10 years | 68 | 24.7 |
| Enterprise type | planting enterprise | 44 | 16.0 |
| | processing enterprise | 20 | 7.3 |
| | retail and wholesale enterprise | 35 | 12.7 |
| | logistics enterprise | 5 | 1.8 |
| | peasant household | 171 | 62.2 |
| Enterprise property | state-owned enterprise | 2 | 0.7 |
| | private enterprise | 52 | 18.9 |
| | limited company | 21 | 7.6 |
| | individual | 200 | 72.7 |

According to Table 1, 24.7% of the companies have been operating for more than 10 years, 32.4% for 5–10 years, and 42.9% for less than 5 years. From the distribution of enterprises, it can be seen that, farmers account for 62.2%, the planting, the retail and wholesale enterprises account for 16.0 and 12.7%, respectively. In terms of the company property, individuals accounted for 72.7% and private enterprises accounted for 18.9%. The number of individuals in the survey grows slightly, and the overall proportion of enterprises is distributed evenly. In general, the survey scope was wide enough, and the proportion of samples was reasonable, indicating these meet the research requirements.

### 3.2. Data Reliability and Validity Analysis

According to the research design, there are 19 observed variables within 4 dimensions included in this questionnaire. Table 2 demonstrates the results of the scale reliability coefficient test.

In Table 2, it showed that all the coeffecent $\alpha$ values were greater than 0.8, and that of the total scale was greater than 0.9, indicating the result data had high reliability and internal consistency, which meet the requirements of modeling and analysis. The Kaiser–Meyer–Olkin (KMO) statistic was further measured, and the result was 0.925, indicating that the correlation between variables was strong. Moreover, the $p$ value calculated by Bartlett's spherical test was significant, indicating that the questionnaire data was suitable for factor analysis.

Varimax rotation of the sample data was performed, as shown in Table 3. Finally, four factors with characteristic roots greater than 1 were extracted, corresponding to four variables respectively. The cumulative variance was 64.613%, suggesting the sample validity was good, which further indicated that the design of the questionnaire was reasonable, the results can reflect the real level of the measured object.

**Table 2.** Reliability and validity analysis of the scale.

| Observation Dimension | Measurement Items | Normalization Factor Load | AVE | CR | Cronbach's $\alpha$ (Coefficient $\alpha$) | The Population Value of Coefficient $\alpha$ |
|---|---|---|---|---|---|---|
| OC model scale | The company has established a completed online and offline system to implement cross-channel linkage sales of Chinese Wolfberry. | 0.665 | 0.591 | 0.852 | 0.850 | |
| | The company can automatically distribute orders according to the customer's address, and allows dealers in a certain region to support delivery services of Chinese Wolfberry. | 0.637 | | | | |
| | The company is capable to integrate new e-commerce technologies into existing facilities rapidly and effectively. | 0.734 | | | | 0.928 |
| | The company's e-commerce staff is competent in managing and maintaining various data. | 0.675 | | | | |
| SSC collaboration scale | The company and business partners share marketing information. | 0.673 | 0.453 | 0.800 | 0.800 | |
| | The company shares information on product demand forecasts with partners. | 0.649 | | | | |
| | The company shares production schedules and logistical information with partners. | 0.647 | | | | |
| | The company adjusts and improves its production system to meet the product requirements of partners. | 0.616 | | | | |
| | When the economic environment has changed, the company and partners will adapt to new technology solutions in a timely manner. | 0.667 | | | | |
| SSC relationship quality scale | We trust the information provided by cooperative partners. | 0.674 | 0.504 | 0.859 | 0.858 | |
| | We trust that our cooperative partners abide by signed contracts. | 0.625 | | | | |
| | Relationships with partners are worth maintaining as far as possible. | 0.663 | | | | |
| | We rely on supports from partners. | 0.599 | | | | |
| | We believe that partners are crucial to us to succeed in our bushiness. | 0.649 | | | | |
| | We need partners to assist us achieve goals. | 0.602 | | | | |
| Service innovation scale | Collaboration has enabled the company to complete service concept innovation. | 0.705 | 0.602 | 0.858 | 0.858 | |
| | Collaboration has enabled the company to promote service technology innovation. | 0.680 | | | | |
| | Collaboration has enabled the company to develop service process innovation. | 0.711 | | | | |
| | Collaboration has enabled the company to innovate service offerings. | 0.655 | | | | |

**Table 3.** Total variance of factor interpretation.

| Ingredients | Initial Eigenvalue | | | Extract Sum of Squares Load | | | Rotation Sum of Squares Loading | | |
|---|---|---|---|---|---|---|---|---|---|
| | Total | % of Variance | Cumulative% | Total | % of Variance | Cumulative% | Total | % of Variance | Cumulative% |
| 1 | 8.281 | 43.583 | 43.583 | 8.281 | 43.583 | 43.583 | 3.721 | 19.583 | 19.583 |
| 2 | 1.651 | 8.688 | 52.271 | 1.651 | 8.688 | 52.271 | 3.511 | 18.480 | 38.062 |
| 3 | 1.334 | 7.020 | 59.291 | 1.334 | 7.020 | 59.291 | 2.892 | 15.221 | 53.283 |
| 4 | 1.011 | 5.322 | 64.613 | 1.011 | 5.322 | 64.613 | 2.153 | 11.330 | 64.613 |

### 3.3. Correlation Analysis of Variables

The correlation coefficient analyses are shown in Table 4. The OC model was positively correlated with SSC collaboration and service innovation, with coefficients of 0.651 and 0.586, respectively. There was a positive correlation between SSC collaboration and service innovation (coefficient: 0.653), suggesting that the questionnaire variables had good correlation, which laid a good foundation for the hypothesis test.

**Table 4.** Correlation between OC model, SSC collaboration and service innovation.

| | Mean Value | Standard Deviation | OC Model | SSC Collaboration | SCRQ | Service Innovation |
|---|---|---|---|---|---|---|
| OC model | 3.832 | 0.776 | 1 | | | |
| SSC collaboration | 3.896 | 0.659 | 0.651 ** | 1 | | |
| SCRQ | 3.921 | 0.673 | 0.530 ** | 0.616 ** | 1 | |
| Service Innovation | 3.990 | 0.677 | 0.586 ** | 0.653 ** | 0.533 ** | 1 |

**: At the level of 0.01 (double tailed test), the correlation is significant.

## 4. Results and Discussions

### 4.1. Analysis of Model Fitting Effect

Based on the above analysis, the AMOS22.0 software was used to estimate the model in Figure 1. The test results are shown in Table 5.

**Table 5.** Fitting effect of the model.

| Indicators | Evaluation Criterion | | Fitted Value |
|---|---|---|---|
| | Acceptable | Good | |
| $\chi^2/\mathrm{df}$ | (3.0, 5.0) | <3.0 | 2.863 |
| GFI | (0.7, 0.9) | >0.9 | 0.906 |
| AGFI | (0.7, 0.9) | >0.9 | 0.861 |
| RMR | (0.05, 0.1) | <0.05 | 0.041 |
| RMSEA | (0.05, 0.1) | <0.05 | 0.082 |
| CFI | (0.7, 0.9) | >0.9 | 0.934 |
| NFI | (0.7, 0.9) | >0.9 | 0.902 |
| TLI | (0.7, 0.9) | >0.9 | 0.916 |
| IFI | (0.7, 0.9) | >0.9 | 0.934 |
| RFI | (0.7, 0.9) | >0.9 | 0.877 |

The results of Table 5 shows that the value of $\chi^2/\mathrm{df}$ was 2.863, GFI was 0.906, RMSEA was 0.082, all met the requirements. The values of CFI, NFI, TLI and IFI were greater than 0.9, indicating a strong fitting effect. The fitting result of the model was good.

### 4.2. Analysis of Hypothesis Results

4.2.1. Analysis of the Hypothesis Test

The data in Table 5 were further analyzed, and the results were obtained in Table 6.

**Table 6.** Hypothesis test results.

| ID | Path | Path Coefficient | Scalar Estimation | Critical Ratio | Significance Level | Hypothesis | Results |
|---|---|---|---|---|---|---|---|
| 1 | SSC collaboration <--- OC model | 0.870 | 0.093 | 9.335 | *** | H1 | support |
| 2 | Service innovation <--- OC mode | 0.226 | 0.110 | 2.050 | * | H2 | support |
| 3 | Service innovation <--- SSC Collaboration | 0.559 | 0.107 | 5.227 | *** | H3 | support |

Notes: *: *p* value was significant at the level of 0.05; ***: *p* value was significant at the 0.001.

The results of Table 6 shows that: the OC model had a significant positive impact on collaboration SSC collaboration (path coefficient: 0.870, $p < 0.001$). The OC model had a significant positive impact on service innovation (path coefficient: 0.226, path coefficient: 0.226, *p* value was significant at the level of 0.05). SSC collaboration had a significant positive impact on service innovation (path coefficient was 0.559, $p < 0.001$). The hypotheses H1, H2 and H3 were supported. The results of Tables 7 and 8 shows the intermediary model results of SSC collaboration.

**Table 7.** Intermediary model test of SSC collaboration.

| Variable | Supply Chain Collaboration | | Service Innovation | | Service Innovation | |
|---|---|---|---|---|---|---|
| | $\beta$ | $t$ | $\beta$ | $t$ | $\beta$ | $t$ |
| OC | 0.0390 | 14.1648 *** | 0.0507 | 4.8173 *** | 0.0428 | 11.9610 *** |
| SC collaboration | | | 0.0597 | 8.1223 *** | | |
| $R^2$ | 0.4236 | | 0.4719 | | 0.3439 | |
| F | 200.6416 *** | | 121.5418 *** | | 143.0648 *** | |

Notes: All variables were normalized with regression equation, the same below. ***: $p$ value was significant at the 0.001.

**Table 8.** Breakdown of total effect, direct effect and intermediary effect.

| Items | Effect | BootSE | BootLLCI | BootULCI | Proportion (%) |
|---|---|---|---|---|---|
| Total effect | 0.5119 | 0.0480 | 0.4170 | 0.6060 | |
| Direct effect | 0.2441 | 0.0557 | 0.1350 | 0.3556 | 47.69% |
| Intermediary effect of SCC | 0.2678 | 0.0466 | 0.1795 | 0.3637 | 52.31% |

Note: SCC: Supply chain collaboration.

The Model 4 (a simple intermediary model) in macro of SPSS compiled by Hayes [43] was used to test the intermediary effect. The results showed that (Tables 7 and 8), the OC model had a significant impact on service innovation ($\beta = 0.0428$, $t = 11.9610$, $p < 0.01$), and when the intermediary variable was added, the direct impact of the OC model on service innovation was still significant ($\beta = 0.0507$, $t = 4.8173$, $p < 0.01$). The OC model had a significant impact on SSC collaboration ($\beta = 0.0390$, $t = 14.1648$, $p < 0.01$), and the impact of SSC collaboration on service innovation was also significant ($\beta = 0.0597$, $t = 8.1223$, $p < 0.01$), indicating the SSC collaboration played a partial intermediary effect between OC model and service innovation. In addition, the upper and lower limits of the bootstrap at 95% confidence interval. The direct and indirect effects were 0.2441 and 0.2678, respectively. The intermediary model holds.

4.2.2. Analysis of Relationship Quality Regulation Effect in SSC

Added the SSC relationship quality to the model, the results were shown in Table 9.

In this paper, we have used the Process v3.4 plug-in of SPSS24.0 and referenced Hayes' Model 7 (a mediating model with the first half regulated, consistent with the theoretical model in this study) to perform a supervisory effect analysis. To ensure the operability of the analysis and the accuracy of the analytical results, the impact of control variables such as firm uptime, firm type and firm nature is considered prior to conducting the moderating effect analysis in SPSS. The results showed that (Table 9), after putting the SC relationship quality into the model, the interaction term of OC model and SSC relationship quality has a significant effect on SSC collaboration ($\beta = 0.0398$, $t = 2.6126$, $p < 0.01$), indicating that the relationship quality of SSC can regulate the impact of OC model on SSC collaboration.

**Table 9.** Test of mediating effect model with regulation.

| Variates | Supply Chain Collaboration | | Service Innovation | |
|---|---|---|---|---|
| | $\beta$ | $t$ | $\beta$ | $t$ |
| OC model | 0.1592 | −0.1173 | 0.0507 | 4.8173 *** |
| SC relationship quality | 0.1573 | −0.1435 | | |
| SC collaboration | | | 0.0597 | 8.1223 *** |
| OC model × SC relationship quality | 0.0398 | 2.6126 *** | | |
| $R^2$ | 0.5373 | | 0.4719 | |
| F | 104.8995 *** | | 121.5418 *** | |

Notes: ***: $p$ value was significant at 0.001.

Further analysis showed that (Table 10, Figure 2), the OC model had a small negative impact on service innovation with a low level of SSC relationship quality (M-1SD), while for the subjects with a high SSC relationship quality (M + 1SD), the OC model had a significant positive impact on service innovation. This indicated that, with the improvement level of SSC relationship quality, the OC model on service innovation was gradually increasing (Table 10).

**Table 10.** Direct and intermediary effects of supply chain relationship quality at different levels.

|  | SSC Relationship Quality | Effect | BootSE | BootLLCI | BootULCI |
|---|---|---|---|---|---|
| Direct effect | eff1 (M-1SD) | 0.1546 | 0.0371 | 0.084 | 0.2293 |
|  | eff2 (M) | 0.1885 | 0.0367 | 0.1221 | 0.2661 |
|  | eff3 (M + 1SD) | 0.2224 | 0.0439 | 0.1460 | 0.3171 |
| Intermediary effect of SSC coordination | eff2-eff1 | 0.0339 | 0.0174 | 0.0090 | 0.0766 |
|  | eff3-eff1 | 0.0678 | 0.0349 | 0.0179 | 0.1531 |
|  | eff3-eff2 | 0.0339 | 0.0174 | 0.0090 | 0.0766 |

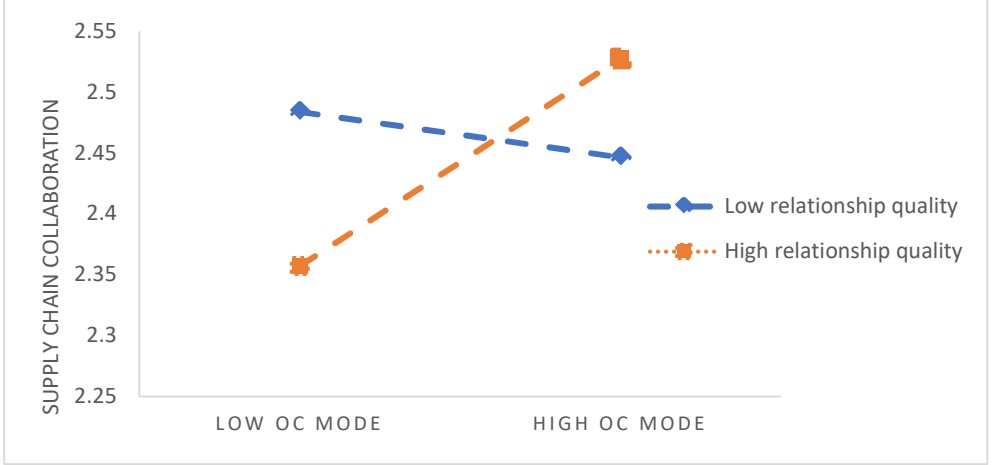

**Figure 2.** The regulating role of SC relationship quality between OC model and SC collaboration.

In addition, at the three levels of SSC relationship quality, the intermediary effect of SSC collaboration in the relationship between OC model and service innovation showed a trend of first increasing and then decreasing (Table 10). With the improvement of SSC relationship quality, OC model can promote its service innovation by improving SSC collaboration to a certain extent, and the hypothesis H4 was verified.

*4.3. Discussion of Study Results*

From the results of this study, the OC model had a significant positive impact on SSC collaboration. The path coefficient of OC model and SSC collaboration was 0.870 ($p < 0.001$), It shows that the OC model with online and offline integration is established to promote information sharing and increase customer shopping experience and product distribution efficiency. The OC model promotes service innovation by sharing market transaction information and promoting mutual collaboration among SSC actors.

Moreover, the OC model of agricultural products effectively promotes enterprise service innovation. Empirical research showed that the OC model had a significant positive impact on service innovation. The path coefficient of OC model and service innovation was 0.226 ($p < 0.05$), enterprises build online and offline OC sales model, through online communication with customers to provide information on product quality, product traceability, and product usage; offline sales enhance customer satisfaction by providing services such

as delivery and after-sales service. This information communication achieves innovation in terms of service concept, technology, process and content.

In addition, agricultural SSC collaboration reflected the collaboration and information sharing among the main bodies of the SC, which together had a significant positive impact on service innovation. The results show that the path coefficient of SSC collaboration and service innovation is 0.559 ($p < 0.001$), which indicates that a close partnership with SC enterprises can promote continuous innovation in enterprise SSC. In order to cooperate better and more efficiently, enterprises need to integrate and transform the existing operation model and adjust the cooperation model in time and transform the production system to continuously adapt to environmental changes and achieve service innovation.

From Tables 9 and 10, it can be seen that relationship quality significantly and positively regulates the effect of OC model on SSC collaboration. The higher the quality of SC relationship in agricultural services, the stronger the effect of OC model on SSC collaboration will be. The study suggests that the effect of OC model on SSC collaboration is enhanced by establishing trusting relationships, discussing issues openly and sharing benefits with partners over time. Therefore, in the context of SSC relationship quality, enterprises should pay attention to the relationship with partners to improve SSC relationship quality.

## 5. Conclusions

In the process of China's economy turning to high-quality development, consumer demand has changed from focusing on product quality to the quality of SC services. Thus, building a stable quality, collaborative and efficient SSC has become a strategic choice for the marketing of agricultural products in Western China. This paper showed that the agricultural SSC collaboration under the OC model is the key to promote the agricultural development and showed a significant positive impact on SSC collaboration. Enterprises should build an OC business platform and strengthen the information communication ability of online and offline channels actively to promote agricultural SSC collaboration. Moreover, the OC model can effectively promote service innovation. The OC model based on the modern Internet, Internet of things, big data and related leading technologies will enhance the competitiveness of enterprises and full experience service innovation with needs of online and offline groups of consumers. Agricultural SSC collaboration reflects the mutual collaboration and information sharing among the main bodies of the SC, which together have a significant positive impact on service innovation. Enterprises can promote service innovation to enhance marketing competitiveness through strategic collaboration, cultural collaboration, business collaboration, information collaboration and other ways. In addition, the quality of SC relationship can regulate the OC mode and SSC coordination. Thus, enterprises should build a good relationship with SC members and improve the relationship quality with trust and commitment.

The management implications of this study include: Firstly, this study finds the important value of OC model for agricultural products sales and service innovation. Therefore, the relevant personnel of enterprises should actively study its operating mechanism and path, realize the unified management of information on commodities, orders, payments and customers through online and offline integration, in order to achieve the integration of information flow, capital flow and logistics, and provide necessary conditions for realizing enterprise service innovation. Secondly, the SSC collaboration of agricultural products has an important impact on service innovation. Enterprises should enhance the innovation of organizational structure, reform management mechanism, establish the connection of SSC subjects by sharing information, establishing synergistic relationships actively, and promote the efficiency of internal organizational integration and external organizational coordination. Thirdly, the positive impact of relationship quality on service innovation provides impetus to enhance the relationship of agricultural SSC. Enterprises should develop the ability of communication and coordination ability among agricultural SSC subjects, establish trust relationship, improve relationship quality, and promote product sales and service innovation actively.

Although the findings of the study have value, there are still limitations. Chinese wolfberry is one of the special agricultural products in western China, with a relatively significant planting area and market sales. The research of this product is representative. Chinese Wolfberry products are not only sold offline, but also sold online, with huge OC sales. The main relationship of Wolfberry's and SSC's is clear, which can reflect the synergistic relationship within each other. However, this paper selects only one Wolfberry agricultural product for research and investigates the cross-sectional data of enterprises. More products can be involved in future research to reveal the synergistic relationship of the SSC of agricultural products in western China, and jointly promote service innovation. Secondly, this study measured the relationship quality of SSC subjects in agricultural products enterprises to reveal the influence of relationship quality on agricultural products SSC collaboration. However, the study did not give a further discussion about how to improve the relationship quality, how to lead to a better SSC operation efficiency, and how to enhance the competitiveness of agricultural products market in western China.

**Author Contributions:** Conceptualization, B.Y. (Baojun Yang) and S.S.; methodology, T.M. and B.Y. (Bo Yuan); software, N.Y. and Y.L.; formal analysis, B.Y. (Baojun Yang) and T.M.; resources, B.Y. (Baojun Yang) and J.L.; writing—original draft preparation, B.Y. (Baojun Yang) and B.Y. (Bo Yuan); writing—review and editing, S.S. and J.L.; supervision, B.Y. (Baojun Yang); data curation, R.J. and Y.W. All authors have read and agreed to the published version of the manuscript.

**Funding:** This research was supported by the Social Science Foundation of Ningxia Province with Grant No.21NXAGL02.

**Institutional Review Board Statement:** Not applicable.

**Informed Consent Statement:** Not applicable.

**Data Availability Statement:** The data presented in this study are available upon request from the corresponding author.

**Acknowledgments:** The work is partially supported by the Center of Econometrics, Chiang Mai University, and the China-ASEAN High-Quality Development Research Center at Shandong University of Finance and Economics.

**Conflicts of Interest:** The authors declare no conflict of interest.

## Abbreviations

Supply chain coordination (SCC); supply chain (SC); service supply chain (SSC); Product service supply chain (PSSC); supply chain management (SCM); Dual-channel (DC); Omni-channel (OC).

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
