# Peer review of "Effect of Relationship Quality in Collaboration and Innovation of Agricultural Service Supply Chain under Omni-Channel Model"

_agriculture, doi:10.3390/agriculture12111932_

Round 1

Reviewer 1 Report

-        Please recheck the paper for possible grammatical errors.

-        In introduction section page 2, second paragraph, line 8, please correct DAN…it seems the year of publication missing, Dan (?)!

-        It is better you name section to as literature background/ review. Following section under 2.2 could be your section 3 and outlined as Research methodology. Name section 3 as section 4 and five conclusion.

-        I would like to see your Questionnaire and loading factor analysis to be inserted in the text.

-        As you named AMOS, prerequisite for using it is that you need to do Normalizing test but we can’t see any. It seems you just used AMOS for good of fitness. Beside I think there is a problem for moderator variable analysis in this research. While we have SEM tools which is stronger than SPSS, how it comes to test your moderator variable by SPSS. For designing a model and testing it PLS would be better choice.

-        Before conclusions, managerial implications of work is missing. Please strength this section.

Reviewer 2 Report

The paper is generally well written, although it requires another edit for some errors. For example, l59 has an incomplete sentence, as does lines 236-237, as well as l287. The authors should use less passive language, for example, l247 would be improved if it simple said "Table 2 demonstrates....".

The paper is well argued and clearly structured. However, it is somewhat limited by focusing on one product. Conclusions are also a bit weak and could be improved by a bit more detail and tying together main arguments and hypohtesis from the paper. It would also benefit from addressing limitations more precisely. 

Reviewer 3 Report

The paper is well written and clear. Research questions are well defined and the result well presented.

Regarding the state of the art inside the Section "introduction", I suggest to include also previous results regarding the introduction of Industry 4.0 technologies in order to increase the trust, the perceived quality between producers and customers. These are in fact services (e.g. the introduction of NFC tag able to guarantee the origiliaty of the product).

After the sentence "Chamhuri and Batt studied the store selection behavior of customers when purchasing agricultural products and found that customers who purchased in modern retail stores placed more importance on services such as convenience and entertainment" I suggest to insert the following reference:

- Bandinelli R., Fani V., Rinaldi R., Customer acceptance of NFC technology: An exploratory study in the wine industry (2017) International Journal of RF Technologies: Research and Applications, 8 (1-2), pp. 1 - 16

Figure 1: please remove the red line under Omni-channel
